# Efficient self-organization of informal public transport networks

Kush Mohan Mittal[1], Marc Timme ®[1,2] & Malte Schröder ®[1] ✉

The Global South, encompassing more than 80% of the world population, heavily relies on informal paratransit services with ad-hoc routes. Yet, it remains unclear how efficiently such informal public transport services organize and operate. Here, we analyze and compare the structural efficiency of more than 7000 formal and informal bus service routes in 36 cities across 22 countries globally. Intriguingly, informal transport self-organizes in ways at or above efficiency levels of centralized services. They exhibit fewer detours, more uniform paths, and comparable interconnectivities, all while remaining profitable without the major subsidies common in the Global North. These insights challenge the global perception of informal transport as an inferior alternative to centrally organized services. More generally, analyzing large-scale microscopic transport data and condensing them into informative macroscopic observables may qualitatively improve system understanding and reveal specific options to create more accessible, efficient, and sustainable public transport solutions.

Human mobility essentially underlies modern societies[1–4]. Public transport services, in particular, enable social and economic participation by providing accessible mobility for the majority of the population[5–7]. At the same time, they support sustainable urban mobility by providing low-emission transport for a large number of travelers while requiring less space and resources than private motor vehicles[7–10]. Efficiently designing and operating public transport systems thus constitutes a cornerstone for solving current challenges for sustainable and livable cities worldwide[6,9].

Most cities in developed countries of the Global North provide public transport as a centrally organized service, with trains, subways, trams, and buses following predetermined lines and fixed schedules[11,12]. Despite their benefits of offering a high degree of predictability and availability, for example, by operating also in off-peak hours[13], public transport is often not sufficiently attractive to compete with the flexibility and ease-of-use of private motor vehicles[14,15]. Consequently, public transport in the Global North consistently makes up a smaller fraction of the modal split than private motor vehicles[15–17].

In contrast, the majority of developing countries in the Global South, encompassing over 80% of the world's population[18], rely on informal, publicly accessible transportation to fulfill daily mobility needs[19–22]. Centrally organized public transport is often not available at all or remains prohibitively expensive for the majority of the population. Instead, a collection of informal, privately operated services provides demand-driven transportation, typically by (mini)buses[21–24].

However, their informal nature has traditionally hindered access to these systems and their operational data such that research so far was limited to small scales, either being focused on individual case studies, limited to specific regions, or both[24–27]. In particular, how such informal systems organize their routes and how efficient they may be, especially compared to formal public transport in the Global North, is not well understood to date.

Here we analyze on a global scale the overall structural properties of more than 7000 bus routes, systematically comparing routes emerging in informal transport (mini)bus services with routes of centrally organized line services. We find that route networks in informal transport of the Global South often self-organize highly efficiently, outperforming centrally planned routes in the Global North. These insights challenge existing paradigms and may overthrow the common global practice of replacing informal by centrally organized line services and thereby pave the way to enabling more sustainable and broadly accessible public transport solutions worldwide.

[1]Chair of Network Dynamics, Institute of Theoretical Physics and Center for Advancing Electronics Dresden (cfaed), TUD Dresden University of Technology, Dresden, Germany. [2]Lakeside Labs, Klagenfurt, Austria. ✉e-mail: malte.schroeder@tu-dresden.de

## Results

### Informal transport services

Unorganized, informal transport describes a broad class of operations offering various forms of for-profit mobility services, often provided by private individuals. Informal transport may be broadly classified into two types with distinct service schemes. The first includes fully dynamic, often highly unregulated taxi-like services with small vehicles such as auto-rickshaws and motorbikes roaming around in search of passengers with dynamic, unplanned pickups and drop-offs[21,23,28]. The second covers larger-scale services providing shared transportation for multiple passengers with mini-buses and vans with roughly fixed service corridors but usually without fixed, predefined stops or schedules[21,23,28–30]. In the following, we focus on the second type of service comparable to bus public transport. We specifically differentiate informal bus services without strict schedules or fixed stops (in the Global South) from formal services with a fixed, predetermined route, fixed stop locations, and a fixed schedule (in the Global North).

While the routes of informal services are not fixed, they are usually restricted to corridors with some regular major stops (e.g., major exchanges or hubs)[24,25]. The corridors of the routes are often agreed upon between competing drivers or groups of service providers, and drivers of informal services typically adhere to these corridors to avoid conflict with competitors or fines from traffic police[25,31]. However, even when exact routes are specified in operating licenses, drivers in practice often adjust their route and stop locations dynamically, for example, to find more customers, circumvent traffic, or avoid police checkpoints when they receive relevant information from passing drivers[21,23,24]. In this way, new

routes may emerge and grow organically and will often be licensed a-posteriori as demand increases and drivers begin serving the route directly when it becomes profitable[21,24,25,30]. The overall route networks of informal services thus self-organize largely without central planning.

Simultaneously, with little or no oversight, profit maximization of drivers and service providers often results in bad vehicle conditions and drivers exceeding the maximum occupancy of their vehicles[21,30]. Moreover, while drivers typically offer their service with some regularity (e.g. when demand and expected profits are high in the morning), service is not guaranteed, waiting times may be long, or stops may be skipped by drivers, resulting in low reliability and potentially low personal safety compared to formal services[20,22,23]. Despite these limitations, informal services are often the primary mode of transport for the majority of the population in developing regions and help establish social equality by providing mobility to all.

### Self-organization of informal transport routes

As a consequence of the self-organization of informal services, reliable digital data about these systems have traditionally been severely limited. Recently, however, various public and private initiatives have made increasing efforts to map informal transport routes in order to improve the accessibility and reliability of these services[27,32,33].

To reveal fundamental structural features of the informal transport networks, we systematically analyze route data made available on OpenStreetMap (see Supplementary Note 1 including Supplementary Table 1). A typical data set for a city contains GPS tracks of buses sampled at discrete points (Fig. 1a, b) and additional information such as an

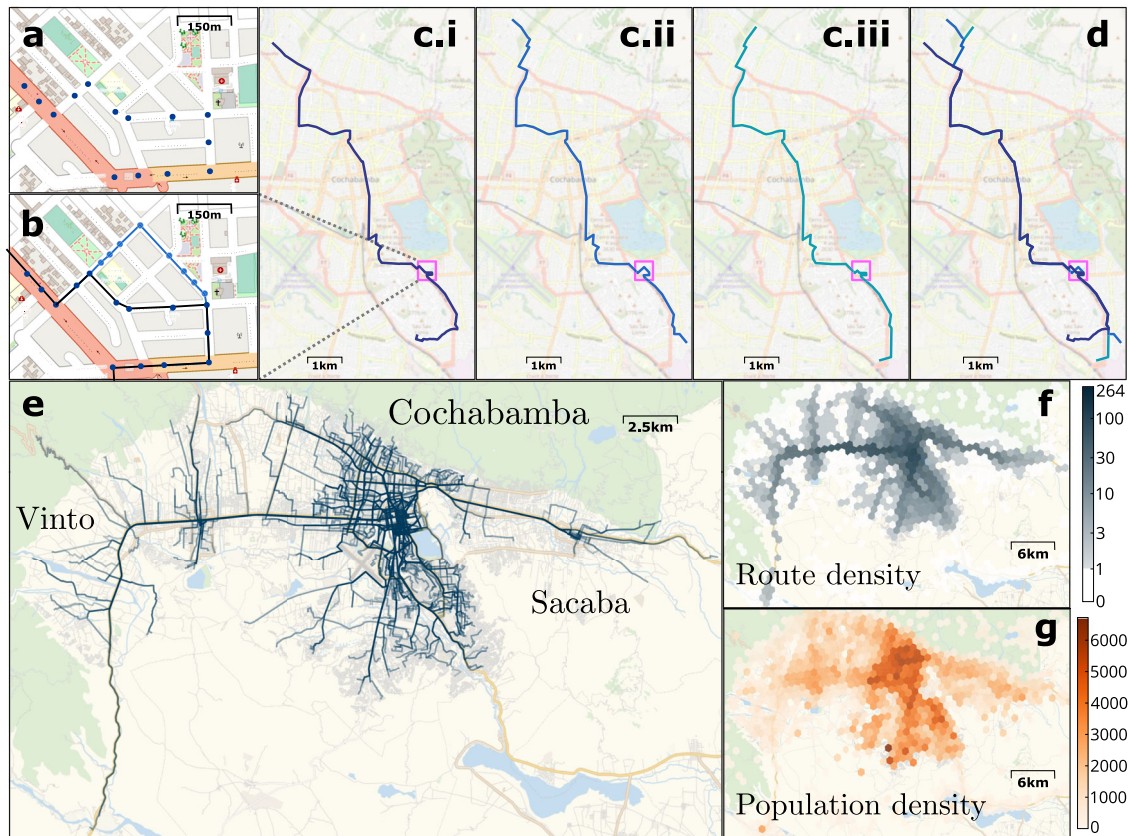

**Fig. 1 | Self-organization of routes in informal public transport networks.** Illustration of the data for the urban region of Cochabamba, Bolivia, South America. **a** A bus route is determined from data as the trajectory driven by a bus, spatially sampled as a sequence of GPS positions (see "Methods"). **b** Different routes belonging to the same ID vary from each other only locally. **c.i**–**c.iii**, **d** Several variations of the route self-organize into consistent lines with high overlap. The three routes only differ at the beginning and the end as well as at a small localized

part highlighted by pink rectangles in panels (**c**) and (**d**) shown in more detail in panels (**a**) and (**b**). **e** The collection of all 431 tracked routes with 130 different IDs covers the whole city of Cochabamba. **f**, **g** Counting the number of routes crossing 400m-sized hexagonal areas of the city estimates the service density that resembles the population density, a rough proxy for the local demand (see Supplementary Note 3 for details).

ID identifying the bus or (formal transport) line operated on. We interpret each trajectory from the data as a single, individual bus route (Fig. 1c). Informal bus routes belonging to the same ID (e.g., routes of the same bus) consistently follow the same paths with only minor deviations, confirming that informal transport routes largely adhere to fixed corridors similar to fixed lines in formalized services (Fig. 1c, d and Supplementary Note 2). Different routes of the same ID differ, for instance, on return trips due to one-way streets; as the drivers adapt to demand and adjust routes dynamically to visit additional stops; or to avoid traffic congestion along certain streets depending on the time of day. Despite these highly dynamic boundary conditions, the collection of all routes self-organizes such that it covers the entire city area and reflects the distribution of the local population density that may be interpreted as a rough proxy for local demand (see Fig. 1e–g and Supplementary Note 3). Figure 1 illustrates the self-organization of individual routes and route networks for the city of Cochabamba, Bolivia.

## Quantifying structural features of routes

The structural properties of the routes driven by the buses essentially underlie the quality of service and especially the efficiency of both formal and informal public transport systems. While some detours compared to the shortest paths between stops may be required to cover a given service area or even result in faster trips by avoiding highly congested areas, smaller detours in general improve service quality by reducing actual and perceived travel times, making public transport more attractive compared to other mobility options[14,34–37].

The detour profile

$$d_x = \frac{\ell_x - s_x}{\ell_x}, \tag{1}$$

captures the microscopic structural properties of a route by quantifying the fraction of the length $\ell_x$ of a small route segment at relative position $x \in [0, 1]$ along the route spent on detours compared to the length $s_x$ of the shortest path between the start and the end point of that segment in the underlying street network (Fig. 2a–d, see "Methods" and Supplementary Note 2 for details). Here we choose segments with length $\ell_x$ as close to $\Delta L = L/3$ as possible such that there are three non-overlapping segments in a route with $x = 0$ labeling the first, $x = 0.5$ the middle, and $x = 1$ the last third of the route. The choice of the segment length $\Delta L = L/3$ provides easily interpretable observables by clearly attributing detours

to non-overlapping segments at the beginning, middle, and end of the route while still encompassing the whole route. Shorter segments increase the local resolution of the detour profile but decrease the overall detours observed, longer segments increase the overlap of segments, making it more difficult to attribute features of the detour profile to specific parts of the route. The results are robust with respect to changes in segment lengths $\Delta L$ relative to the total length of the route as well as for fixed-length segments, see Supplementary Notes 4 and 5.

The detour profile $d_x$ reflects the non-straightness of the route relative to the shortest path and quantitatively captures the visual differences in the structure of the route along its length. The example profile displayed in Fig. 2d illustrates strongly heterogeneous detour fractions along the route (compare Fig. 2a, see Fig. 2e for a schematic illustration): detours are small in the middle of the route with the lengths of the segments and the shortest paths deviating by only around 5% whereas detours are much larger and deviations reach above 20% towards both ends of the route. Interestingly, such strong detour heterogeneity similarly occurs for most routes in Cochabamba.

The median detour profile $\langle d_x \rangle$ of all bus routes in a city quantifies the overall variation of detours along routes in that city as well as the degree of heterogeneity of the detours along the routes (see Fig. 2f). The features of detour profiles may substantially vary among individual routes. A detailed analysis of the individual routes in Cochabamba reveals a large diversity of route structures (see Supplementary Notes 4 and 5 for additional examples), but no strong correlations with the city structure or, for example, the type of street used by the route (see Supplementary Notes 6 and 7). Despite these large variations, median detour profiles consistently capture the typical structure of routes and are characteristic of a given city (see Supplementary Note 10 for examples from more cities).

## Structural efficiency across cities

Two basic observables summarize the information contained in the detour profile of a route and quantitatively characterize its structure. The detour heterogeneity

$$\xi = \frac{d_0 - 2\,d_{0.5} + d_1}{2}, \tag{2}$$

defined by the coarse-grained second derivative of the detour profile, captures the variability of the detour fraction between the beginning

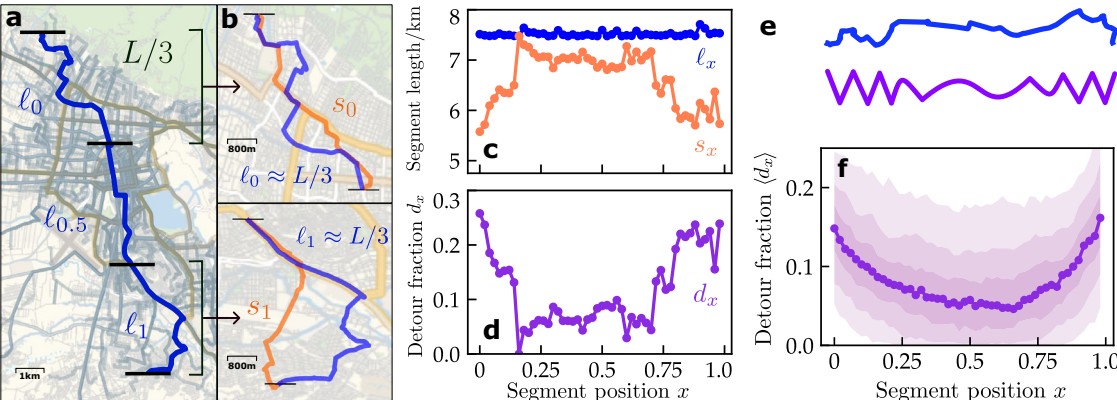

**Fig. 2 | Detour profiles characterize route structure. a** One route (blue solid line) of length $L$ is partitioned into segments of length $\ell_x \approx \Delta L = L/3, x \in [0, 1]$. Three non-overlapping segments at $x \in \{0, 0.5, 1\}$ are illustrated with their ends marked by black bars. **b** The first and last segments of lengths $\ell_0$ and $\ell_1$ (blue) compared to their respective shortest paths of lengths $s_0$ and $s_1$ (orange). **c** Length $\ell_x$ of segments (blue) and length $s_x$ of the corresponding shortest paths between segment ends (orange) across segment position $x$ along the route shown in panel (**a**). **d** Resulting detour profile of the route given by the detour fraction $d_x$ as a function

of position $x$ along that route (see "Methods" and Supplementary Note 2 for details). **e** Schematic illustration of the geometric interpretation of the detour profile, showing the actual route (blue) and a sketch (purple) with a qualitatively similar detour profile. **f** The median detour profile $\langle d_x \rangle$ across all routes in Cochabamba (compare Fig. 1c) captures the overall detour profile characteristic for that city (see Supplementary Note 10 for example for more cities). The 40–60%, 30–70%, and 20–80% percentiles are marked by increasingly light shading.

and end compared to the middle of a route. Large positive values of the detour heterogeneity $\xi$ imply a convex detour profile and longer detours toward the beginning and end of a route compared to the middle. Conversely, large negative values of $\xi$ imply a concave detour profile with shorter detours towards the beginning and end of a route.

The total detour

$$D = d_0 + d_{0.5} + d_1, \qquad (3)$$

defined as the sum of the three detour fractions, measures the overall efficiency of the route structure. The total detour specifically distinguishes routes with small heterogeneity $\xi$ that may either be straight with respect to the street network topology and closely follow shortest paths (small $D$) or exhibit large detours everywhere along the route (large $D$). Conversely, a large detour heterogeneity $\xi$ along the route always also implies a large total detour $D$. This relation between the two observables is clearest in the limiting case of routes with zero detours along their middle segment, $d_{0.5} = 0$, where $D = 2\xi$ (see "Methods" and Supplementary Notes 2 and 8 for more details on the interpretation of these observables).

Public transport routes in most cities exhibit positive median detour heterogeneity $\langle \xi \rangle$ (Fig. 3a). In principle, high detour heterogeneity may result from the universal geometric constraints that routes crossing a city have to cover a larger area towards the outskirts as the radial distance from the city center increases and the density of routes decreases (see Supplementary Note 9 for a minimal model demonstrating this effect). Intriguingly, however, our analysis reveals a consistent difference between cities with different types of public

transport: many cities with informal public transport exhibit lower detour heterogeneity $\langle \xi \rangle$ than cities with centrally organized public transport (Fig. 3a, see also Supplementary Note 10 for detour profiles of various sample cities). This pattern is consistent with an analysis of the individual routes in different cities (see Supplementary Note 11 for additional scatter and contour density plots of the distribution of routes in $\xi$-$D$-space) and confirmed by an analysis of the rank statistics of $\langle \xi \rangle$ across cities, strongly suggesting that the median $\langle \xi \rangle$ of the detour heterogeneity is systematically lower for informal services than it is for formal ones (see Table 1 and Supplementary Note 13). Strong detour heterogeneity and convex detour profiles pose additional inconvenience at the beginning and end of the routes, often to commuters traveling to and from the outskirts of a city, adding to their already long travel times due to the distance covered and affecting them more than those traveling closer to city centers[38-41].

Routes of informal transport also frequently exhibit lower total median detour $\langle D \rangle$ than routes of centrally organized services (Fig. 3b), again confirmed by an analysis of the rank statistics of the median total detour $\langle D \rangle$ across all cities (see Table 1 and Supplementary Note 13 for more details). Overall, the self-organized routes of informal public transport services thus demonstrate the tendency to be structurally more efficient than those of centrally organized services (Fig. 3c–f, see also Supplementary Note 11 for a more detailed analysis of the structure of the individual routes in sample cities).

We similarly find that the structural efficiency of self-organized informal services is often larger than that of centrally organized ones for other related properties of the route networks, for instance, the population-weighted detour heterogeneity and the population-

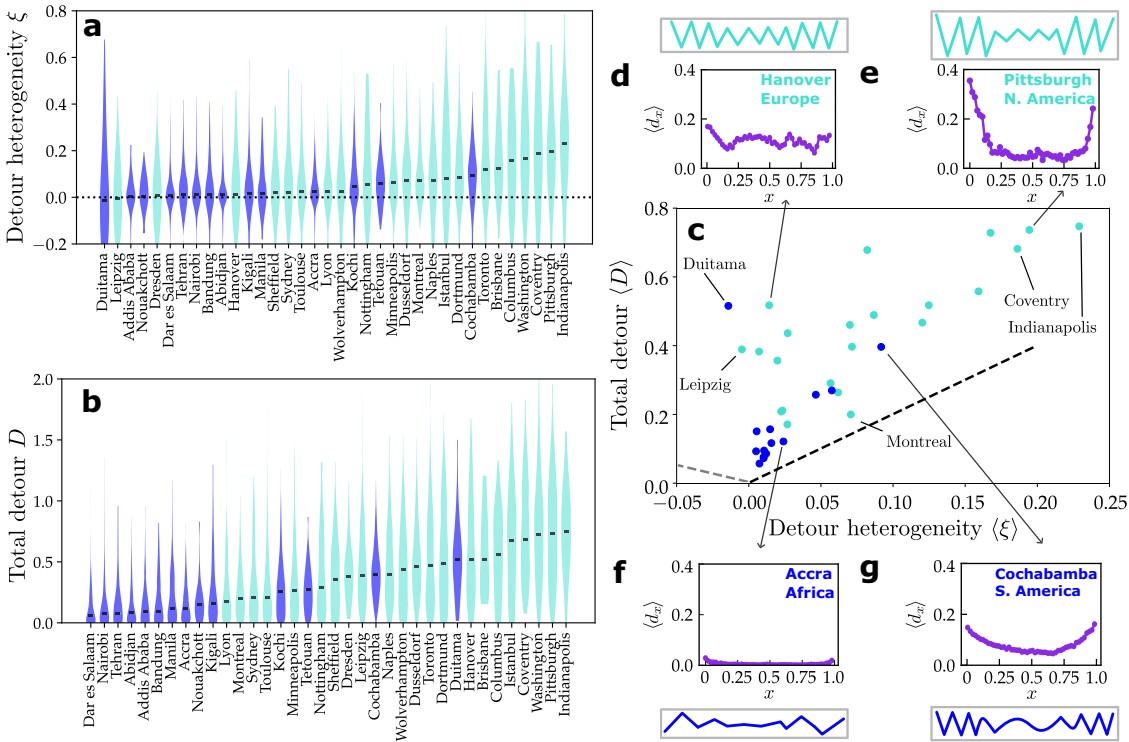

**Fig. 3 | Efficient routes of informal public transport. a** Distribution of detour heterogeneities $\xi$ for each city in a violin plot (the widths of the distributions are not comparable across cities). Cities with informal (dark blue) and formalized (light blue) public transport are ranked in increasing order according to their median detour heterogeneity $\langle \xi \rangle$ (black dash). Cities with informal services consistently exhibit lower detour heterogeneity than cities with formal transport (see Table 1). **b** Distribution of total detours $D$. Cities with informal (dark blue) and formalized (light blue) public transport are ranked in increasing order according to their median total detour $\langle D \rangle$ (black dash). Cities with informal services consistently exhibit lower total detours than cities with formal services (see Table 1). **c** Observables for all cities plotted in the parameter space of $\langle \xi \rangle$ and $\langle D \rangle$ indicate the clear tendency of informal services (dark blue) to frequently exhibit both lower heterogeneities and lower total detours. Intriguingly, most cities accumulate close to the boundary line $\xi \leq D/2$ (gray dashed line), indicating that the middle segments of routes consistently have little detour across cities (see "Methods" and Supplementary Note 8 for more details). **d–g** Detour profiles of four selected cities with schematic sketches of corresponding route structures.

**Table 1 | High structural efficiency of informal services**

| Observable | Median Informal | Median Formal | p-value |
|---|---|---|---|
| Detour Heterogeneity $\xi^*$ | 0.011 | 0.070 | 0.0007 |
| Total Detour $D^*$ | 0.12 | 0.45 | 0.00005 |
| Number of intermediate routes $C$ | 0.670 | 0.675 | 0.48 |

Both detour heterogeneity $\xi$ and total detour $D$ exhibit substantially lower median values (median across city medians) for cities with informal services than they do for cities with formal services. The $p$-values of the Mann-Whitney U rank statistics test indicate the probability of a similar or more extreme outcome of the ordering if the observations from both categories were drawn from the same distribution, thus strongly suggesting that the observed differences are non-random (indicated by a*, compare Fig. 3a, b). In contrast, the average number $C$ of intermediate routes required to switch between two routes of informal services is statistically indistinguishable from that of formal services. Population-weighted comparisons support the same conclusions (see Supplementary Note 13 and Supplementary Table 3).

weighted total detour (see Supplementary Note 13). Furthermore, despite fewer detours, the number $C$ of intermediate routes required to switch between any two routes in informal transport systems are as small as in formalized services (see "Methods" and Supplementary Note 12).

All these results remain robust with respect to changes in the segment length $\Delta L$ and other details of the analysis (see Supplementary Notes 4, 5 and 13). Importantly, these results are not explained by the number of individual routes nor the length or density of routes in a city (see Supplementary Note 1 including Supplementary Table 1).

## Discussion

Efficient structural properties of public transport routes are a fundamental prerequisite for a viable service. The above results indicate that in informal public transport systems, most bus routes self-organize highly efficiently across regions and independently of the specific geographic or infrastructural features of the cities. Intriguingly, self-organized route structures often are equally or more efficient than those of formal, centrally organized services. In particular, informal routes may exhibit overall lower detours, more homogeneous distributions of detours along the routes, population-adapted route densities, and form networks with a similar number of intermediate routes. Interestingly, the informal services of the Global South achieve these features that underlie their high efficiency without major public transport subsidies common in the Global North[42]. In contrast, they typically need to be inherently profitable to support the operators' livelihood[21,23,24,43].

The observed efficient self-organization of informal route structures may, in part, originate from the dynamic interactions among many drivers and customers in contrast to static services in the Global North. For example, competition between drivers and operators for quality-sensitive customers has been suggested to directly promote efficient, more direct routes[20]. Our global comparative analysis confirms the potential of self-organized transport to form highly efficient services. However, an efficient route structure alone is not sufficient for an efficient service[44,45]. For example, interpreting the interconnectivity of route networks in terms of service quality also requires information on the waiting times and reliability of connections[46]. Similarly, services may not be useful to customers despite direct routes if the frequency of trips is too low to offer reliable transportation[45,47].

Due to the limited availability of data on informal transport services, the above analysis focused only on the spatial properties of the routes for which large-scale microscopic data are available. Ongoing and future data collection and analysis initiatives may help enhance our understanding of how informal transport self-organizes and operates. However, to date, the informal and dynamic nature of these services has severely limited the availability of microscopic data, such

as temporal information on waiting or travel times or time-resolved usage. Key aspects to be addressed with such data need to cover the temporal unreliability of informal services affecting waiting times of customers[24], the availability of these services in terms of operating times in relation to demand[21], as well as technical and social safety issues[22].

Overall, our results challenge the perception of informal transport as an inferior form of public transport and the current global practice of replacing informal transport with traditional, more formal alternatives[21,23,24,48,49]. They call for more detailed, data-driven analyses to underpin the design of efficient and sustainable public transport services that include – not replace – core elements of informal transport. This requires both more detailed observations of temporal and demand data in detailed case studies and across cities as well as theoretical modeling of the evolution of routes on different time scales from driver decisions on individual trips to the evolution of the overall route network. A better understanding of informal transport services and how to effectively utilize their dynamic organization may thus not only initiate a change of perspective on informal transportation services but also help avoid a largely unsustainable transition to car-centric cities in the Global South[50]. Insights on these self-organized systems may furthermore contribute towards mitigating existing car-centric transport in the Global North by pointing towards equitable, efficient, and sustainable public transport solutions.

## Methods
### Data sources
We analyze route data from 36 cities across the globe (see Supplementary Note 1 for details). Cities were selected with the aim of ensuring a balanced representation of formal and informal transport and a wide distribution of geographical locations. We specifically selected medium-sized cities that primarily rely on buses for public transport to be able to better compare formal and informal services. This condition specifically excludes very large cities in developed countries with extensive metro or rail networks. Importantly, the selected cities with formal and informal transport have a comparable density of bus routes, enabling us to make meaningful comparisons across the two categories.

We obtain the route data as GPS tracks from OpenStreetMap and categorize the data for each city as either informal or formal based on the "operator" tag of the route as well as additional information provided by various organizations and individuals who contributed to the data on OpenStreetMap[27,29,32,33].

In our analysis, we further rely on the Kontur population data set[51] to assess population distribution globally. This data set utilizes H3 hexagons with population estimates at a resolution of 400 meters. When computing route densities or population-weighted observables, we compute the number of distinct routes passing through one of the hexagons or, conversely, consider all hexagons a route passes through. To define the total areas and population of a city (see Supplementary Note 1), we consider only those hexagons from the data set where at least one route passes through.

### Route data
We obtain the route data from OpenStreetMap as GPS tracks with an average distance of 30 m between successive points. Importantly, the average gap between GPS points is significantly smaller than the average route length of more than 10 km such that microscopic route details are always visible in the data.

Each route is also tagged with additional information, including, in particular, a unique ID (representing, for example, the bus or the bus line the route belongs to). In the data, multiple routes may belong to the same ID, for example, if a bus goes back and forth between two locations, has different routes depending on the time of day (in formal

services), or drivers slightly vary their routes (in informal services), compare Fig. 1c, d. Data from cities with formal transport shows a median of approximately 1.4 routes per unique bus ID, whereas informal transport shows a median of about 2 routes per bus ID (see also Supplementary Note 2). We analyze each route individually instead of aggregated routes with the same ID to retain information on the actual trips of buses. To avoid over-representing a single route with multiple observations, we present all city-aggregated information as the median over all routes instead of the mean.

## Detour profile

The length $L$ of a route is the sum of distances between the successive GPS points. The GPS data is sufficiently detailed that this captures the influence of street geometry like curved roads. We quantify the structural efficiency of a route of length $L$ by its detour profile $d_x$, denoting the relative detour of smaller route segments of length $L/3$ at position $x \in [0, 1]$ along the route. Each segment starts at a GPS point in the route and extends along the route to the first GPS point such that it has length $\ell_x \geq L/3$. Consequently, $x = 0$ denotes the segment starting at the first point in the route and ending at the first point at or beyond length $L/3$. Similarly, $x = 1$ denotes the segment starting at the last point before or at length $2L/3$ and ending at or slightly before the last point in the route. Due to the small spacing between successive GPS coordinates, the differences in segment length are small compared to the total length of a route (compare also Fig. 2c and Supplementary Note 2).

To compute the detour factor $d_x$ of a segment at position $x$, we compute the shortest path distance $s_x$ between its start and end point in the street network from OpenStreetMap. The detour factor of the segment is then given as the relative detour

$$d_x = \frac{\ell_x - s_x}{\ell_x}$$

normalized with respect to the segment length to keep the normalization the same for all segments since $\ell_x \approx L/3$ (see Supplementary Note 2 for details and interpretation). The detour factor $d_x$ as a function of the position $x$ along the route is the detour profile of the route. The detour profile of a city is the median $\langle d_x \rangle$ over all individual routes in the city for each normalized location $x$ along the routes. In all figures, we illustrate the spatially resolved detour profiles by evaluating them at 50 approximately equally spaced points along the routes.

## Detour heterogeneity and total detour

We quantify the shape of the detour profile by computing aggregated observables from the relative detour of the three non-overlapping segments at positions $x = 0$, $x = 0.5$, and $x = 1$. Specifically, we compute the detour heterogeneity

$$\xi = \frac{d_0 - 2d_{0.5} + d_1}{2} \tag{4}$$

and the total detour

$$D = d_0 + d_{0.5} + d_1, \tag{5}$$

as well as the corresponding medians $\langle \xi \rangle$ and $\langle D \rangle$ across all individual routes in a city. Various combinations of the magnitude of values for detour heterogeneity $\xi$ and total detour $D$ lead to distinct route structures illustrated in Supplementary Note 2.

However, these parameters are not uncorrelated. By definition of the parameters, we have a mathematical limitation for the detour heterogeneity $\xi \leq D/2$ for a given $D$ since

$$\xi = \frac{d_0 + d_1}{2} - d_{0.5} = \frac{d_0 + d_{0.5} + d_1}{2} - \frac{3d_{0.5}}{2} = \frac{D}{2} - \frac{3d_{0.5}}{2} \leq \frac{D}{2}. \tag{6}$$

This limit is reached when the middle segment of a route has zero detour fraction ($d_{0.5} = 0$), and the entire detour is concentrated in the end segments ($d_0, d_1$). A similar boundary $\xi \geq -D$ exists for routes where the detour is concentrated on the middle segment (see Supplementary Note 8 for details). Importantly, this implies that a route with total detour $D = 0$ will also have detour heterogeneity $\xi = 0$.

## Intermediate routes

For any two routes A and B in a city, we define the number $C_{A,B}$ of intermediate routes as the number of other bus routes a customer needs to take to switch from route A to route B[52]. For instance $C_{A,B} = 0$ if a customer can directly switch from route $A$ to route $B$ at some point, and $C_{A,B} = 1$ if a customer needs to take one other bus route in between route A and route B. From this data, we compute the average intermediate routes $C_A = \frac{1}{N_{\text{routes}} - 1} \sum_B C_{AB}$ of a route $A$ by averaging over all other routes $B$. The city characteristic $C$ is then given by the median of $C_A$ across all routes in the city (see Supplementary Note 12 for details).

## Reporting summary

Further information on research design is available in the Nature Portfolio Reporting Summary linked to this article.

## Data availability

The route data is openly available on OpenStreetMap[53]. Additional metadata (such as whether routes are formal or informal) are available and have been accumulated from specific contributors[27,29,32,33]. Snapshots of the data are available at[54].

## Code availability

The commented code is available at[55].

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

## Acknowledgements

The authors thank David Storch, Mark Zuidgeest, and Obiora Nnene for insightful discussions. K.M.M. acknowledges support from the German

Academic Exchange Service (Deutscher Akademischer Austauschdienst, DAAD). M.T. acknowledges support from the German Research Foundation (Deutsche Forschungsgemeinschaft, DFG) through the Center for Advancing Electronics Dresden (cfaed). M.T. and M.S. acknowledge support from the German Federal Ministry for Digital and Transport (BMDV) as part of the innovation initiative mFund under grant number 19F1155A.

## Author contributions

K.M.M. initiated the research with help from M.S. and M.T. All authors conceived and designed the research and provided methods and analysis tools. K.M.M. collected and analyzed the empirical data with help from M.S. All authors contributed to evaluating and interpreting the results and wrote the manuscript.

## Funding

## Competing interests

The authors declare no competing interests.
