## [Peer Review File · Nature Communications]

REVIEWER COMMENTS

Reviewer #1 (Remarks to the Author):

This paper aims to analyze and compare the efficiency of formal and informal bus service routes. A metric named detour profile is proposed to quantify the gap between actual and shortest routes. Based on the detour profile, the authors further propose two metrics, detour heterogeneity and total detour, to describe the structural efficiency of street networks across cities.

The motivation behind this work is valuable; however, the undertaken analysis can not effectively assess the efficiency of public transportation services. The following are some points of concern:

(1) The paper lacks a clear definitions of formal and informal bus service routes. It is uncertain whether both routes have fixed bus stops and whether the informal routes have the flexibility to park at any location.

(2) When discussing service efficiency, the paper lacks an analysis of whether detours are prompted by travel demand or traffic conditions. Public transportation services aim to satisfy more passengers with genuine needs and ensure shorter travel times. However, the detour profile is defined with the shortest path, which may not accurately describe efficiency. Therefore, it is suggested to consider selecting a route that has shorter travel time and meets more needs.

(3) The strategy of dividing the routes into three equal segments is perplexing. Why not simply use the bus stations or parking spots instead? Heterogeneity could also be defined based on entropy. Moreover, the relationship between Figure 2c, d and the segmentation strategy of $L/3$ is unclear. Since the route is divided into three segments, why there are so many points (50) in the figure. Similarly, in Figure S7, why does the number of points remain the same when selecting different segmentation lengths ($L/5$ to $L/2$)?

(4) The paper proposes the number of intermediate routes $\$C\$$ to describe the route interconnectivity. However, there are minimal differences in metrics between formal and informal services. How to interpret this phenomenon, and does it suggest that this indicator may be unsuitable for assessing route interconnectivity? Additionally, what does a p-value of 0.48 signify in this case?

(5) It is recommended to maintain consistent color of the base case between Figure S12d-f and Figure S12c.

Reviewer #2 (Remarks to the Author):

The author contends that public transport is essential for societal mobility and economic participation. Analyzing over 7,000 bus routes globally, the study suggests that informal transport systems in the Global South self-organize more efficiently than centrally planned services in the Global North, challenging the common practice of favoring formal services and advocating for more sustainable and accessible public transport solutions worldwide.

Major issues with the paper:

The conclusions drawn by the authors are consistent. City studies are diverse and have different features. However the main metrics used to compare the different cities rely solely on geographical features, and there is no consideration of the street network topology, and the main results are solely descriptive.

Both the detour profiles definition and the detour heterogeneity are interesting and well-constructed; however, these metrics are intrinsically linked to the street network topology, and there is no analysis regarding any network features.

Finally, always in the same direction of the network analysis, it would be interesting to have some understanding of the relation of the detour with the type of street for capacity/type. This data is usually available on OpenStreetMap.

The conclusion of the paper is interesting, although the capacity of moving people around the city is not taken into consideration. Some checks on the quantity of runs per route are needed to validate the main conclusion. Defining a more self-organized system for capillarity or the ability to reach different areas of the city using detour metrics is a very compelling idea. However, how can we be sure that these informal lines are really able to cover the needs of the geographical area instead of only providing an answer with a couple of runs per day per route? I did not see any reference to either capacity or frequency of service, or any control in this direction.

Fig. 3C suggests that there might be some linear relation for some of the cities; it might be interesting to explore the cities in the top/left quadrant and capture the features that provide these two regimes. The considerations in support of the figure in the supplementary materials are limited to a table and a distribution description of the detour metrics. No attempt to disentangle the relations has been made.

The paper provides a new technical framework however there is no validation in terms of the variables that describe the systems. For example checks and cross-checks in terms of the street topology of the cities, or the consideration in terms of population density. The descriptive analysis of the detour metrics is consistent for the detour metrics, but does not study any social-economic aspect that it might drive the demand of the routes (Fig S8 or fig. s6 are only density plot qualitative approach and for only one city). More checks are needed to fully support the main claim of the authors.

Minor issues:

It would be nice to add an introduction and results section in the main paper structure.

Figs. 1.c and d: It would be nice to have a zoom-in on the area of route change.

Figure S14: It might be nice to have a density plot or some metrics for comparison between cities; it is not clear from the figure's statement: "exhibit consistently lower values of D and ξ ."

Fig. 3C: It seems that there might be some linear relation for some of the cities; it might be interesting to explore the cities in the top/left quadrant and capture the features that provide these two regimes.

Reviewer #3 (Remarks to the Author):

The authors assemble public transport route data (on >7 000 individual service routes) from 36 cities across the world, which span cities with both 'centralised' and 'informal' public transport operations. Route data are analysed from the perspective of service route directness, and service route network 'interconnectivity'. They find that informal public transport service routes exhibited fewer detours, and exhibited comparable network interconnectivity, relative to centralised public

transport service routes. On the basis of these findings they conclude that informal public transport services are not as inefficient or inferior as commonly perceived.

Multi-context qualitative analysis of this type is rare, and consequently I think the paper will be of considerable interest to scholars of public transport networks in the Global South.

I have two main comments:

1. The authors state that “informal public transport services are organized and operated on a broadly informal level, with drivers communicating their route choices without central planning or central information and often making ad-hoc changes ... bus service providers actively adapt their routes to serve many customers ...”. This creates an impression that drivers of informal public transport service vehicles (as opposed to informal for-hire services like motorcycle taxis) are at liberty to select their service routes. While this liberty is true for service headways and service spans, this is uncommon for routes. Many, if not most, cities in the Global South have route licensing regimes that stipulate the routes upon which service provision is permitted. Route deviation is not uncommon, but it is of limited spatial extent because straying off the route can attract fines from law enforcement, or worse from competitors.

The degree to which there is public authority control over the assignment of routes within informal public transport networks is not widely understood. In the Sub-Saharan African contexts I am most familiar with, it is generally the case that routes start off unlicensed, and route licenses are awarded reactively. It is probably fair to say that in these contexts the service network is therefore more the result of the operators themselves, than some public authority supply-demand route planning exercise. But this is poorly understood. So I found myself wondering whether the findings in the paper regarding service route directness and detour, are more the product of industry self-organisation, or effective public authority route licensing regimes. The authors reflection on this would be helpful to the reader in interpreting the results.

2. Regarding the reflection on future research needs in the concluding Discussion section, the authors state that the results of the paper “call for more detailed, data-driven analyses to underpin the design of efficient and sustainable public transport services that include – not replace – core elements of informal transport”. I would find it interesting to know where the authors think investigations of this type should be directed. While comparing favourably with centralised public transport service route networks with respect to directness and interconnectivity, informal public transport service route networks have temporal coverage gaps (or market failures), as well as service duplication and operating inefficiencies, that demand policy attention, if these services are to form a core element of future public transport networks. The author’s conclusions on the research priorities that emerge from their findings would be instructive.

Replies to the comments of the referees

on the manuscript

Efficient self-organization of informal public transport networks

by Kush Mohan Mittal, Marc Timme, Malte Schröder

GENERAL REPLIES

We thank all reviewers for their thorough comments and constructive suggestions. All reviewers agree that the manuscript provides 'interesting and well-constructed' measures to address a 'valuable' topic that is rarely discussed. Reviewer 3 highlights that the manuscript 'will be of considerable interest to scholars of public transport networks in the Global South'.

In their comments, the reviewers ask for clarification and additional background information, especially on the operation of informal transport services, to better contextualize the results and their implications. They also wonder about the robustness of the detour measures introduced in the manuscript and their interpretation beyond spatial characteristics of service quality.

In the manuscript, we analyze bus public transport networks across a wide range of cities, specifically comparing the route networks of informal, self-organized services in cities of the Global South to centrally organized route networks of formal public transport services in the Global North. Based on the available data, we develop new measures to quantify the spatial characteristics and detours of the routes and route networks as a fundamental prerequisite underlying efficient transport networks. Our analysis demonstrates the potential efficiency benefits of self-organized routes in informal public transport compared to most centrally planned public transport services. The findings highlight the need to better understand the emergence and self-organization of these properties and the options to improve both informal and formal public transport services.

In the revised manuscript, we have carefully addressed all comments and suggestions of the reviewers. In particular, we have extended the introduction and discussion to provide more context of the operations of informal public transport and more clearly point out the implications of our findings for public transport networks and future research into informal and self-organized public transport. We have also revised the explanation of the analysis methods and results and added further analyses in the Supplementary Information that additionally support the robustness of our findings, as suggested by the reviewers.

We address the individual comments by the reviewers in more detail below.

REPLIES TO THE COMMENTS OF REVIEWER 1

Referee comment:

This paper aims to analyze and compare the efficiency of formal and informal bus service routes. A metric named detour profile is proposed to quantify the gap between actual and shortest routes. Based on the detour profile, the authors further propose two metrics, detour heterogeneity and total detour, to describe the structural efficiency of street networks across cities.

Authors' response:

We thank the reviewer for their concise summary and their detailed comments to help improve the manuscript.

Referee comment:

The motivation behind this work is valuable; however, the undertaken analysis can not effectively assess the efficiency of public transportation services. The following are some points of concern:

(1) The paper lacks a clear definitions of formal and informal bus service routes. It is uncertain whether both routes have fixed bus stops and whether the informal routes have the flexibility to park at any location.

Authors' response:

We thank the reviewer for raising this important point.

A strict definition of formal and informal services is almost impossible due to the varied nature of the operations of (both of) these services. We therefore employ the following working definition:

In general, we regard any service with *a fixed, predetermined route, fixed stop locations, and a fixed schedule* as a formal service. This is the case for all formal public transport services analyzed in the manuscript and in general for most centrally planned public transport services in the Global North. While there also exist demand-driven shuttles and taxis as part of some formal services, they are rare and do not occur in the routes analyzed in the manuscript.

In contrast, we call services *informal* if they do not generally operate with exactly prescribed routes and fixed schedules. The services still mostly adhere to a fixed *corridor* for their trips and drivers will usually visit a number of fixed stops (hubs similar to large public transport interchanges) along their route. These recurring routes are also visible in the

data, illustrated for example in Fig. 1a-d in the manuscript. However, drivers do often stop at unscheduled points along the route to pick up or drop of passengers and typically have the freedom to deviate from their standard routes in search of customers. While the routes of most vehicles are now de-facto fixed, they were not designed top-down by central planing. Instead, they have often emerged from the self-organization between different drivers and operators (by agreement or by conflict over the most profitable routes) and have been fixed *a-posteriori* by the involved parties or licensed by the government (see also the first comment by reviewer 3 and references [24,25,31] in the revised manuscript for a more detailed description of typical operations of informal paratransit services). Informal transport services thus operate in between strictly organized public transport services and fully demand-driven ridehailing services.

Due to the large diversity of the operational procedures between different regions, a full discussion of all details of informal public transport operations would be impossible in the scope of the manuscript. At the same time, it is exactly this diversity that makes our systematic multi-context analysis crucial in understanding the core overarching features of these systems across cities, as also pointed out by reviewer 3.

We have extended the introduction in the revised manuscript to better differentiate the relevant features of formal and informal transport services in the context of our analysis. We have especially focused on clarifying the description of the operation of informal transport and their route networks in contrast to organized public transport in the Global North. Finally, we have more clearly highlighted the relevant references for more in-depth discussions and additional details.

Referee comment:

(2) When discussing service efficiency, the paper lacks an analysis of whether detours are prompted by travel demand or traffic conditions. Public transportation services aim to satisfy more passengers with genuine needs and ensure shorter travel times. However, the detour profile is defined with the shortest path, which may not accurately describe efficiency. Therefore, it is suggested to consider selecting a route that has shorter travel time and meets more needs.

Authors' response:

We thank the reviewer for this valuable comment.

The underlying reasons for detours and deviations from a route are highly diverse. Drivers may decide to drop off (a larger group of) customers closer to their destination. They may decide to take a detour in search of more customers, for example, based on their experience with the trip and the general demand conditions or due to information they receive from other drivers. Drivers may also deviate from their routes to avoid checkpoints (e.g., to avoid fines for overcrowded vehicles) or to avoid traffic (e.g., to more quickly start a new trip to increase their earnings). Attributing different route changes to specific conditions is unfortunately impossible from the available data and the interpretation of such observations may vary strongly across the different locations, depending on local operational schemes and regulations. Moreover, the observation of the route itself may have affected the behavior of the drivers (e.g., to make fewer detours).

We agree with the reviewer that service efficiency requires more than spatial efficiency. However, more accurate data required to assess the service quality in more detail, like the timing of informal transport trips or stop times and additional information on the demand, is not available across cities (largely due to the informal nature of these services). We therefore focus on the spatial properties of the routes as a fundamental prerequisite for an efficient service. We concur with the reviewers that a more detailed analysis would be valuable, as also stated in the discussion of the (revised) manuscript.

We have extended the introduction and explanation of informal transport operations in the revised manuscript to provide additional context for the driver behavior and potential detours not included in the observed routes.

Referee comment:

(3) The strategy of dividing the routes into three equal segments is perplexing. Why not simply use the bus stations or parking spots instead? Heterogeneity could also be defined based on entropy. Moreover, the relationship between Figure 2c, d and the segmentation strategy of $L/3$ is unclear. Since the route is divided into three segments, why there are so many points (50) in the figure. Similarly, in Figure S7, why does the number of points remain the same when selecting different segmentation lengths ($L/5$ to $L/2$)?

Authors' response:

We measure the detour along segments of the route (similar to a sliding window) in order to resolve the spatial structure of the route on a scale that is comparable across different routes and cities and roughly reflects the typical travel distances of customers along parts of the route.

If we were to choose a very fine resolution of the segments (e.g. all stops along a route), we would almost always find no detour as the route between two consecutive stops usually follows the shortest (or fastest) path and these stops are very close together. Moreover, the information on the exact stop locations is not available for most informal transport routes where buses may stop anywhere along their route. On the other hand, if we were to consider the whole route as one, we may find large detours as routes often deviate from the shortest (or fastest) paths in order to cover more area and serve more people, as the reviewer also mentioned in their previous comment.

Together, a high resolution would imply a vanishing signal whereas a maximum signal would provide no spatial resolution of the detour along the route.

The choice of an intermediate segment length $L/3$ relative to the total length L of a route allows us to (1) compare our results across routes with different total lengths and (2) easily attribute the detour to the beginning, middle, and end of a route without overlapping segments (e.g., double counting detours) or missing parts of a route. The results are insensitive to changes in the chosen segment length.

We have revised the Supplementary Information. As now also shown for fixed segment lengths in the new Fig. S8, the qualitative shape of the detour profiles is robust with respect to different choices as long as the segments remain of a sufficiently intermediate length as explained in the previous paragraph.

In the figures illustrating the detour profile, we show the detour fraction for a variety of segments $[aL, (a + 1/3)L]$ along the route with different starting locations $a \in [0, 2/3]$ in terms of the relative distance along the route, normalized to the segment position $x = a/(1 - l) = 3a/2 \in [0, 1]$ shown in the figures (for relative segment length $l = 1/3$). By scanning through the route with different starting locations (as illustrated in Fig. S4 in the Supplementary Information), we obtain the detour profile as a detailed overview of the distribution of the detours along the route. The specific number of 50 points is an arbitrary choice to illustrate the variability along the route. However, since the segments underlying these data points are partially overlapping, we consider only the three non-overlapping segments along the route, $[0, L/3]$, $[L/3, 2L/3]$, and $[2L/3, L]$ at positions $x = 0$, $x = 1/2$, and $x = 1$, respectively, to calculate the aggregated observables D and ξ .

In the revised manuscript, we have more clearly explained the advantages and the interpretation of the segmentation of the routes. Moreover, we have repeated the analysis for fixed absolute segment lengths of 3km, 4km, 5km, and 6km in the revised Supplementary Information, which may more closely reflect typical trip distances of customers of these services at the expense of comparability between different routes. The results further demonstrate the robustness of our analysis, though the direct interpretation and comparison across routes cities is more difficult as lengths of routes and typical trip distances vary with city size or likely even across different routes.

Finally, we thank the reviewer for the intriguing suggestion to quantify the heterogeneity in terms of entropy. While entropy should work as a qualitative measure in principle, we found that certain intricacies arise in interpreting the results. We attempted the following analysis: first, we normalize the detour profile of a route, interpreting it as a probability distribution. Then we compute the entropy S , normalizing it to $\hat{S} \in [0, 1]$ with respect to the minimum and maximum possible value for a distribution defined on 50 data points. Finally, the measure $H = 1 - \hat{S}$ describes the heterogeneity of the detour along the route with $H = 0$ for uniformly distributed detour (maximum entropy) and $H = 1$ for detour only along a single segment (minimum entropy). Multiplying this measure with the total detour D then provides a quantification similar to our detour heterogeneity, which is large only if there are both significant detours along the route and this detour is heterogeneously distributed along the route. For completeness, we include the results for all routes in Cochabamba in the figure below. However, the interpretation of this measure is much more difficult since we cannot easily attribute the detour or heterogeneity to different parts of the routes. We also note again that the detour observation of the segments are not independent since the segments overlap, further complicating the interpretation and normalization of the measure. While this is an interesting approach for future analysis, we did not include it in the Supplementary

Information to avoid confusion.

Referee comment:

(4) *The paper proposes the number of intermediate routes C to describe the route interconnectivity. However, there are minimal differences in metrics between formal and informal services. How to interpret this phenomenon, and does it suggest that this indicator may be unsuitable for assessing route interconnectivity? Additionally, what does a p -value of 0.48 signify in this case?*

Authors' response:

The Mann-Whitney-U-Test employed in the analysis compares the ordering of two classes of random observations to the null hypothesis that there is no statistically discernible difference between the two classes, i.e., values from one class are equally likely to be smaller or larger than values from the other class. The resulting p -value from the test describes how probable it is to find as extreme an ordering of the values from the two classes in terms of the ranks of the individual ordered observations given the null hypothesis (there is no difference between the two classes).

A p -value of 5×10^{-5} as observed for the total detour D and the detour heterogeneity ξ suggests that observations from one class are significantly smaller/larger than the observations from the other class. A similar or more extreme distribution of the observation would only occur with a probability 5×10^{-5} if there was no difference between the two classes.

In contrast, a p -value of 0.48 for the route interconnectivity suggests that there is no discernible ordering between the observations from formal and informal transport. In fact, the p -value of about 0.5 suggests that the ordering is as close to the expected random ordering as possible if both observations were drawn from the same distribution.

The results thus show that there are indeed 'minimal differences in [route connectivity] between formal and informal services', as stated by the reviewer. We note that this identical connectivity measure emerges despite the fact that informal public transport routes are overall straighter with fewer detours. The result thus demonstrates that there is no trade-off between the directness of the routes and the accessibility of different locations (e.g., by requiring many changes), at least in terms of the spatial structure of the routes.

We agree that the total range of the route connectivity is very similar for both types of services, requiring typically 1 to 3 changes to reach any other route. All services form relatively dense networks of routes. An alternative measure for the impact of the interconnectivity of routes may be the total travel time, including time spent waiting for connections. However, this would require currently unavailable information on the timing of trips in informal transport (see also our replies below). Similar analyses have been done in the context of formal transport by the authors of the paper originally suggesting the connectivity measure

(compare Ref. [53] in the main manuscript). As it compares all pairs of routes, the number of intermediate routes also only provides an aggregate, mean-field measure for the interconnectivity of the network. Moreover, a future detailed analysis would also require more details about the specific demand conditions with appropriate data *across cities*.

We have clarified the interpretation of these observations in the discussion in the revised manuscript.

Referee comment:

(5) *It is recommended to maintain consistent color of the base case between Figure S12d-f and Figure S12c.*

Authors' response:

We thank the reviewer for pointing out this source of confusion. We have adjusted the colors of the figures accordingly.

REPLIES TO THE COMMENTS OF REVIEWER 2

Referee comment:

The author contends that public transport is essential for societal mobility and economic participation. Analyzing over 7,000 bus routes globally, the study suggests that informal transport systems in the Global South self-organize more efficiently than centrally planned services in the Global North, challenging the common practice of favoring formal services and advocating for more sustainable and accessible public transport solutions worldwide.

Authors' response:

We thank the reviewer for their concise summary and their detailed comments to help improve the manuscript.

Referee comment:

Major issues with the paper:

The conclusions drawn by the authors are consistent. City studies are diverse and have different features. However the main metrics used to compare the different cities rely solely on geographical features, and there is no consideration of the street network topology, and the main results are solely descriptive.

Both the detour profiles definition and the detour heterogeneity are interesting and well-constructed; however, these metrics are intrinsically linked to the street network topology, and there is no analysis regarding any network features.

Authors' response:

We thank the reviewer for their positive feedback and their suggestion to more closely study the influence of the street network on the route structure.

Our results are based on basic observables representing a summary of the structure of the route networks across cities. The analysis of the route structure naturally relies on the street network topology to define the shortest paths and detours, as also acknowledged by the reviewer. Importantly, there is no direct relation between the route structures and the size of the city or network, as illustrated in Tab. 1 in the Supplementary Information. The observed differences between formal and informal services are not explained simply by the number or density of routes. Moreover, as exemplified for Cochabamba, there is no strong correlation between route structures and city properties. This also holds across the diverse set of cities analyzed. For example, both US cities and South American cities often share a grid-like street network topology, yet still exhibit different route characteristics. Or, Euro-

pean and US cities exhibit similar route characteristics despite their typically very different structure with historic inner cities and pedestrian zones compared to dense downtown areas.

In contrast, there is a clear, statistically significant difference between the structural properties of routes of formal and informal transport. We take these observations to mean that the route structure is determined by the organization of the service more so than by geographic features of the city or street network.

We agree with the reviewer that a mechanistic explanation of the emergent route structures would be highly valuable, especially for the self-organized structure emerging in informal services (compare also the comments of and our replies to reviewer 3). However, we believe that these structures are much more dependent on demand patterns and the interactions between customers and drivers (to change routes to adapt to demand) and different drivers and operators (to fix routes to avoid conflict). We have demonstrated that the routes *do* self-organize, however, how these services self-organize is a major open question and may vary across cities and countries, despite similar underlying driving forces. Unfortunately, the available structural route data are not sufficient to construct reliable models.

In the discussion of the revised manuscript, we have added a brief discussion of additional aspects that may be of interest for future research. Please also see our replies to the following comments for additional analyses and examples.

To avoid potential confusion, we note that we did not fully understand the comment as it mentions that the analysis relies 'solely on geographical features, and there is no consideration of the street network topology' but at the same time acknowledges that the measures introduced 'are intrinsically linked to the street network topology'. We invite the reviewer to clarify their comment in case we did not interpret the intention correctly.

Referee comment:

Finally, always in the same direction of the network analysis, it would be interesting to have some understanding of the relation of the detour with the type of street for capacity/type. This data is usually available on OpenStreetMap.

Authors' response:

We thank the reviewer for this interesting suggestion.

We have performed an analysis of the correlation of the detour along a segment with the street type used by the route segment as defined in OpenStreetMap (highway, primary, secondary, tertiary, residential). While information from OpenStreetMap is often less reliable for developing countries, information is indeed available. However, we note that the street types are often used in different ways across countries (and sometimes even inconsistently within a country). We have thus focused on Cochabamba as the primary example in the manuscript and Supplementary Information for consistent presentation.

The analysis, summarized in Tab. S2 in the revised Supplementary Information, reveals that detours are almost equally distributed across the street types. Detours occur significantly less for routes along highways (trunk), likely due to these segments of a route covering large distances with fewer intermediate stops. In contrast, detours occur slightly more often along trips through smaller streets (secondary). Detours do not occur significantly more often for residential roads than expected, likely due to the abundance of this street type, thus averaging out potential deviations. The results are consistent with our previous analysis of the location of detours along a route within the city center or on the outskirts of the city.

We have added this analysis together with a brief discussion in the revised Supplementary Information and a short note in the revised manuscript.

Referee comment:

The conclusion of the paper is interesting, although the capacity of moving people around the city is not taken into consideration. Some checks on the number of runs per route are needed to validate the main conclusion. Defining a more self-organized system for capillarity or the ability to reach different areas of the city using detour metrics is a very compelling idea. However, how can we be sure that these informal lines are really able to cover the needs of the geographical area instead of only providing an answer with a couple of runs per day per route? I did not see any reference to either capacity or frequency of service, or any control in this direction.

Authors' response:

We thank the reviewer for mentioning this important point.

As already discussed in our replies to Reviewer 1, we agree that overall service quality cannot be judged by the route network alone. However, the structure of the route network is a fundamental prerequisite for an efficient service. Unfortunately, additional data on the frequency of trips, actual travel times, or stop times is typically not available across cities, due to the informal nature of these services. Without fixed schedules, drivers may decide to wait at stops for a long time for more customers or simply decide to not offer any further trips if they have met their earning goal for the day. Similarly, there are other concerns emerging from the typical operation of these services such as personal and traffic safety and overcrowded vehicles. In the manuscript, we focus on the available data to analyze the spatial properties of the route networks as a fundamental prerequisite for an efficient service and low travel times compared to possible alternative options, especially in developed countries where private cars may be both more convenient and significantly faster.

Designing a demand-driven system inspired by the efficient route networks of informal services would need to address the shortcomings of the fully self-organized system in terms of temporal accessibility and reliability. Possible options for this (some also being tested in an attempt to reform informal services) may include regulated, fixed contracts for drivers paid by the hour to ensure services are available instead of effectively self-employed drivers that may stop offering trips at any time or, where possible, more use of digital information processing to enable drivers to respond to demand changes (instead of relying on information passed between drivers on the road), see Ref. [21-24] in the revised manuscript. Our results provide the basis for such future work, highlighting in the first place that some aspects of these self-organized services may be more efficient than those of centrally planned services.

In the revised manuscript, we have more clearly pointed out the scope of the study and clarified the discussion of the limitations with respect to additional (especially temporal) properties required for efficient public transport networks.

Referee comment:

Fig. 3C suggests that there might be some linear relation for some of the cities; it might be interesting to explore the cities in the top/left quadrant and capture the features that provide these two regimes. The considerations in support of the figure in the Supplementary Information are limited to a table and a distribution description of the detour metrics. No attempt to disentangle the relations has been made.

Authors' response:

We thank the reviewer for highlighting this important relationship.

There is indeed a relation between the detour heterogeneity ξ and the total detour D . In particular, a high detour heterogeneity ξ requires some detours along the route and thus some minimum total detour D . Mathematically, these conditions are captured by the inequalities

$$\xi \leq \frac{D}{2} \quad \text{and} \quad -\xi \geq D$$

for positive and negative detour heterogeneity, respectively (compare also Methods in the revised manuscript and Note 8 and Note 11 in the revised Supplementary Information). These constraints hold both for individual routes (see Fig. S12 and S15 in the revised Supplementary Information) and for the aggregated data (see Fig. 3c in the revised manuscript).

Most cities and most routes in these cities exhibit a large, positive detour heterogeneity close to the upper bound $D/2$ (this is also clearly visible in the added density plots in Fig. S16 and S17 in the revised Supplementary Information). This means all of the detours are occurring at either end of the route and (almost) no detour is occurring in the middle segment, as illustrated by the typical U-shape of the detour profile. In contrast, cities in the upper left corner of Fig. 3c (i.e. low absolute value of the detour heterogeneity but high total detour) exhibit a flat detour profile. These are predominantly European cities with formal public transport networks Hannover, Leipzig, Dresden (all Germany), Sheffield, and Wolverhampton (UK) as well as Duitama, (Colombia). Detours in the middle segment of the routes, typically in the city center, may be more likely to emerge here due to routing restrictions in or around old town centers and pedestrian areas with comparatively narrow streets for buses. However, such restrictions also exist in other cities with different detour profiles, and it does not explain the route structure of informal transport in Duitama, Colombia, making it difficult to attribute the route structure to a single, specific aspect of the city structure.

We have more prominently highlighted this relation and its effect on the interpretation of the results in the revised manuscript and the revised Supplementary Information.

Referee comment:

The paper provides a new technical framework however there is no validation in terms of the variables that describe the systems. For example checks and cross-checks in terms of the street topology of the cities, or the consideration in terms of population density. The descriptive analysis of the detour metrics is consistent for the detour metrics, but does not study any social-economic aspect that it might drive the demand of the routes (Fig S8 or fig. s6 are only density plot qualitative approach and for only one city). More checks are needed to fully support the main claim of the authors.

Authors' response:

We thank the reviewer for raising this point.

As described in the previous replies, we focus on the spatial structure of the route network as a fundamental prerequisite for an efficient service. Unfortunately, there are in general not sufficient data on the demand and use of informal transport services across cities. While data like passenger numbers or origin-destination relations exist for (some) centrally organized services, they are missing for the (largely analog) informal services. We show the comparison of the route density and the population density as a simple indicator that the routes (1) cover the populated area of the city and (2) provide better coverage in more densely populated areas. Importantly, even trip demand that is not directly covered by a route is easily met by customers switching as the route interconnectivity in informal services remains high (the number C of intermediate routes is low). While this does not necessarily mean that travel times are minimal due to waiting times and stop times, it provides the basis for an efficient service.

We have included additional analyses of the influence of the street types on the detour factor of the route segments in the revised Supplementary Information (see Tab. S2). We have also further demonstrated the robustness of the suggested detour metrics as requested also by Reviewer 1 (see Fig. S8). To avoid confusion, we have also carefully revised the manuscript to clearly point out the scope of the study and the limitations with respect to temporal aspects of informal public transport services.

Referee comment:

Minor issues:

It would be nice to add an introduction and results section in the main paper structure.

Authors' response:

We have added section headings to provide a clearer structure to the revised manuscript.

Referee comment:

Figs. 1.c and d: It would be nice to have a zoom-in on the area of route change.

Authors' response:

Panels a and b of the figure show the zoomed-in area of the route changes. We have clarified this in the revised manuscript.

Referee comment:

Figure S14: It might be nice to have a density plot or some metrics for comparison between cities; it is not clear from the figure's statement: "exhibit consistently lower values of D and ξ ."

Authors' response:

We thank the reviewer for this suggestion.

The large number of routes per city makes it difficult to identify individual routes and may make it difficult to compare different cities. We have added contour plots of (smoothed) densities directly comparing the cities in the revised Supplementary Information, see the new Figs. S16 and S17. These figures show both a predominant alignment along the limiting line of high detour heterogeneity (see our reply to a previous comment) and a significantly larger spread of total detour and detour heterogeneity for cities with formal transport (the main result previously only shown in Fig. 3 in the manuscript).

Referee comment:

Fig. 3C: It seems that there might be some linear relation for some of the cities; it might be interesting to explore the cities in the top/left quadrant and capture the features that provide these two regimes.

Authors' response:

As described in our reply above and also explained in the revised Supplementary Information, the two measures D and ξ are indeed related. It is not possible to have a high detour heterogeneity without having a total high detour D . We have more clearly explained this relationship and its effect on the interpretation of the results in the revised manuscript and Supplementary Information (for a more detailed reply, see the similar comment above).

REPLIES TO THE COMMENTS OF REVIEWER 3

Referee comment:

The authors assemble public transport route data (on >7 000 individual service routes) from 36 cities across the world, which span cities with both ‘centralised’ and ‘informal’ public transport operations. Route data are analysed from the perspective of service route directness, and service route network ‘interconnectivity’. They find that informal public transport service routes exhibited fewer detours, and exhibited comparable network interconnectivity, relative to centralised public transport service routes. On the basis of these findings they conclude that informal public transport services are not as inefficient or inferior as commonly perceived.

Multi-context qualitative analysis of this type is rare, and consequently I think the paper will be of considerable interest to scholars of public transport networks in the Global South.

Authors’ response:

We thank the reviewer for their concise summary and their positive recommendation of the manuscript.

Referee comment:

I have two main comments: 1. The authors state that “informal public transport services are organized and operated on a broadly informal level, with drivers communicating their route choices without central planning or central information and often making ad-hoc changes . . . bus service providers actively adapt their routes to serve many customers . . .”. This creates an impression that drivers of informal public transport service vehicles (as opposed to informal for-hire services like motorcycle taxis) are at liberty to select their service routes. While this liberty is true for service headways and service spans, this is uncommon for routes. Many, if not most, cities in the Global South have route licensing regimes that stipulate the routes upon which service provision is permitted. Route deviation is not uncommon, but it is of limited spatial extent because straying off the route can attract fines from law enforcement, or worse from competitors.

Authors’ response:

We thank the reviewer for pointing out this inaccurate description of informal public transport operations.

The description we provided was meant to specifically highlight the differences between services with and without central organization of fixed routes and schedules. We did not mean

to imply that the routes are entirely random or may be different every day (which would also make it impossible for customers to navigate the system). Some degree of permanent organization would be necessary even without government licensing or agreements between competing service providers. The fact that the routes of informal transport services remain relatively well-defined and deviations are small is also illustrated by the data (compare Fig. 1a-d in the revised manuscript and Fig. S2 and S3 in the Supplementary Information, indicating more than 70% overlap of similar routes). In fact, the existence of stable routes is a major prerequisite for our analysis of the spatial structure of the route networks.

We more clearly point out this fundamental observation and the underlying organizational structures in the revised manuscript and have extended the explanation of the operations of informal transit, also in line with requests from the other reviewers. We now more carefully distinguish between the general idea of self-organized transport, the differences to centrally organized services, and the typical operational norms in informal public transport. We focus our discussion on the route networks relevant to the present analysis and have more clearly highlighted relevant references for in-depth discussions on additional details beyond the scope of the manuscript.

Referee comment:

The degree to which there is public authority control over the assignment of routes within informal public transport networks is not widely understood. In the Sub-Saharan African contexts I am most familiar with, it is generally the case that routes start off unlicensed, and route licenses are awarded reactively. It is probably fair to say that in these contexts the service network is therefore more the result of the operators themselves, than some public authority supply-demand route planning exercise. But this is poorly understood. So I found myself wondering whether the findings in the paper regarding service route directness and detour, are more the product of industry self-organisation, or effective public authority route licensing regimes. The authors reflection on this would be helpful to the reader in interpreting the results.

Authors' response:

We thank the reviewer for raising this important question.

As described in the previous comment by the reviewer, licensing of routes or agreements between competitors on the allowed service regions often occurs after routes have been established by drivers and operators. Similarly, high demand for trips may incentivize drivers to take risks in establishing a new route, despite the potential negative consequences. It thus seems likely that the route network first emerges from a self-organized process and routes are 'fixed' at some later time. Understanding how this process happens, how dynamic the network remains, and how these routes may change (or not change) in the future is crucial for improving these services as cities grow and develop. The comparatively large agency of drivers and operators suggests that routes may adapt more easily (compared to centrally organized services), for example by discontinuing service when a route is no longer profitable. Moreover, insights into this self-organization process may also help potentially adapt the mechanisms to design better public transport networks in developed countries.

In our opinion, this self-organization of the informal transport routes makes the results from the present manuscript even more surprising. One might naively expect central (global) optimization to generate more efficient route structures whereas self-organized (local) optimization is typically more likely to settle into local optima and inefficient route structures, similar to the differences between Nash-equilibria and social optima in game theoretic problems. The fact that this is not the case highlights (1) the ability of public transport systems to self-organize efficiently, at least in terms of the spatial network structure, and (2) potential shortcomings of rigidly scheduled public transport services (e.g. due to constraints such as providing reliable transportation for most of the day).

We have extended the discussion to more prominently present the importance of the self-organization of the spatial structures in informal transport services and the gap in research

in understanding their evolution, highlighting the potential implications (see also our reply to the next comment).

Referee comment:

2. Regarding the reflection on future research needs in the concluding Discussion section, the authors state that the results of the paper “call for more detailed, data-driven analyses to underpin the design of efficient and sustainable public transport services that include – not replace – core elements of informal transport”. I would find it interesting to know where the authors think investigations of this type should be directed. While comparing favourably with centralised public transport service route networks with respect to directness and inter-connectivity, informal public transport service route networks have temporal coverage gaps (or market failures), as well as service duplication and operating inefficiencies, that demand policy attention, if these services are to form a core element of future public transport networks. The author’s conclusions on the research priorities that emerge from their findings would be instructive.

Authors’ response:

We thank the reviewer for this interesting question.

We agree that informal transport services are likely lacking in other aspects like reliability and safety when compared to centrally organized services. As mentioned in our previous replies (also to the other reviewers), understanding these systems and their shortcomings is crucial to harness the positive features of self-organization.

The most immediate step should likely be to study (and if possible compare) other aspects of efficiency, especially in terms of waiting times, travel times, and reliability of services, ideally in relation to demand, where such data is available or can be easily sourced. Such investigations may provide insights into the trade-offs made in the self-organization of these services and would provide the basis for theoretical modeling to explore what changes or interventions to the self-organization might improve the system without disrupting its ability to adapt to changes in demand. In general, all of these investigations require both additional empirical observations to validate and challenge theoretical models as well as theoretical models to predict and evaluate the effect of new or different interactions and restrictions on these services.

We have slightly extended the discussion in the revised manuscript to include this perspective on future research in the context of our findings.

REVIEWERS' COMMENTS

Reviewer #1 (Remarks to the Author):

This study examines the structural efficiency of bus service routes by analyzing deviations from the shortest paths. The analysis reveals that informal transport services exhibit fewer detours compared to formal ones. Nevertheless, this observation does not substantiate the assertion that informal transport services hold greater value than formal counterparts.

(1) The shortest path does not always represent the most optimal route selection. Furthermore, some detours are meaningful, such as avoiding traffic congestion or serving more passengers. Therefore, solely analyzing detour situations may not provide sufficient insight.

(2) What implications does the reduction in detours hold? What guidance can the findings in the paper provide for urban planners or transportation managers? The impact of the findings in this article is limited, along with its contribution to the research domain.

(3) The article compares detour situations in different cities, concluding that informal bus services exhibit lower detours, indicating higher structural efficiency. However, this evidence is insufficient, as factors like city size and population may influence structural efficiency. It is recommended to conduct comparative analyses between cities with similar urban structures, considering both global southern and northern cities.

Reviewer #2 (Remarks to the Author):

The revised manuscript has undergone significant improvements. The authors have diligently outlined the limitations of the study and offered viable solutions to the questions and issues raised in the previous version. The newly included supplementary figures enhance comprehension of the developed concepts and framework.

MINOR:

However, there are some minor formatting issues and discrepancies in the new manuscript. Formulas 2, 5, and 6 are presented as if they are equations, yet they are merely definitions and lack

explicit delineation as such. These discrepancies should be addressed to ensure clarity and accuracy.

Reviewer #3 (Remarks to the Author):

I am satisfied that the authors have addressed my comments systematically.

I have just one suggested minor revision, as follows.

In their response to my earlier review, the authors agreed with my comment about route establishment and variation. To avoid confusion on the new point they are making in this regard, I suggest the following rewording (changes in uppercase):

The corridors of the routes are often agreed upon between competing drivers or groups of service providers, and drivers of informal services typically adhere to these corridors to avoid conflict with competitors OR FINES FROM TRAFFIC POLICE. However, EVEN WHEN EXACT ROUTES ARE SPECIFIED IN OPERATING LICENSES, IN PRACTICE drivers (are in principle free to-DELETE) OFTEN adjust their route and stop locations, for example, to find more customers, circumvent traffic, or avoid police checkpoints when they receive relevant information from passing drivers.

Reason for rewording: in instances where route alignments are specified in operating licenses, it is not correct to say that the drivers are 'in principle free to adjust routes'.

Replies to the comments of the referees

on the manuscript

Efficient self-organization of informal public transport networks

by Kush Mohan Mittal, Marc Timme, Malte Schröder

REPLIES TO THE COMMENTS OF REVIEWER 1

Referee comment:

This study examines the structural efficiency of bus service routes by analyzing deviations from the shortest paths. The analysis reveals that informal transport services exhibit fewer detours compared to formal ones. Nevertheless, this observation does not substantiate the assertion that informal transport services hold greater value than formal counterparts.

(1) The shortest path does not always represent the most optimal route selection. Furthermore, some detours are meaningful, such as avoiding traffic congestion or serving more passengers. Therefore, solely analyzing detour situations may not provide sufficient insight.

Authors' response:

We thank the reviewer for this valuable comment.

The service quality of public transport is determined by various aspects of the route structure, travel times, and operating scheme. We discuss these additional aspects in the manuscript. However, while detours may not be the only determining factor for optimal routes, detours are a crucial aspect of the structural efficiency of public transport routes. They fundamentally determine travel times and reducing detours is key to enhancing the perception and appeal of public transport, highlighting the importance of selecting the shortest paths to boost ridership. Importantly, our results also demonstrate that detours are not necessarily required to serve more passengers or (dynamically) avoid congestion, evidenced by most informal transport routes exhibiting low detours as drivers attempt to maximize their profit (i.e. the number of passengers they transport per unit time).

Gathering more data on the organization of informal transport routes is crucial for understanding informal transport systems and ultimately enabling more detailed comparisons to formal and optimal transport routes along different, non-structural aspects of service quality. We have clarified the scope of our results and the potential implications for additional aspects of service quality in the revised manuscript.

Referee comment:

(2) What implications does the reduction in detours hold? What guidance can the findings in the paper provide for urban planners or transportation managers? The impact of the findings in this article is limited, along with its contribution to the research domain.

Authors' response:

As stated in the earlier comment and referenced in the main article, detours play a pivotal role in the efficiency and appeal of public transport routes. Our results highlight the need for a thorough, data-driven evaluation of informal transport systems' service quality before considering replacing them with formal systems. Moreover, the findings suggest that informal transport may offer better overall service quality without relying on subsidies typical for formal transport. This indicates that future mobility solutions should potentially incorporate aspects of informal transport's operational principles.

Due to the current lack of detailed data and the large diversity of informal transport operations, detailed analysis and modeling of the organizing principles are currently still missing in the literature. Our results highlight the potential of informal, self-organized public transport. Therefore, we believe that such analyses could be highly valuable in the future to understand the efficient self-organization of public transport networks. We highlight these implications in the revised manuscript.

Referee comment:

(3) The article compares detour situations in different cities, concluding that informal bus services exhibit lower detours, indicating higher structural efficiency. However, this evidence is insufficient, as factors like city size and population may influence structural efficiency. It is recommended to conduct comparative analyses between cities with similar urban structures, considering both global southern and northern cities.

Authors' response:

We thank the reviewer for this valuable comment.

We agree with the reviewer that differences between cities in terms of size, geography, or population play a crucial role in shaping the structure of transport networks in general. Detailed modeling of transport networks requires considering the individual features of the specific city. In contrast, our results provide an overarching, systematic comparison across cities to identify general features of (informal) transport networks.

Moreover, we do discuss the potential relation of the efficiency of the route networks to the city structure, see also Supplementary Note 1, for example, comparing the area covered by the routes and the population they serve. This comparison demonstrates that the observed effects are not directly associated with the topology of the city or simply a larger number of routes. Instead, the observed efficiency of informal transport routes emerges *despite* similar city characteristics.

Finally, as also demonstrated in the manuscript, we observe efficient organization of informal public transport route *across* a wide range of different cities. This is a further indication that underlying organizational features between formal and informal transport are responsible for the observed differences regardless of specific city characteristics.

We have revised the presentation in the manuscript to clarify these implications of the results.

REPLIES TO THE COMMENTS OF REVIEWER 2

Referee comment:

The revised manuscript has undergone significant improvements. The authors have diligently outlined the limitations of the study and offered viable solutions to the questions and issues raised in the previous version. The newly included supplementary figures enhance comprehension of the developed concepts and framework.

MINOR:

However, there are some minor formatting issues and discrepancies in the new manuscript. Formulas 2, 5, and 6 are presented as if they are equations, yet they are merely definitions and lack explicit delineation as such. These discrepancies should be addressed to ensure clarity and accuracy.

Authors' response:

We thank the reviewer for their positive evaluation of the revised manuscript and their suggestion to further improve readability. We have removed the formulas and integrated the notation directly into the text to avoid confusion.

REPLIES TO THE COMMENTS OF REVIEWER 3

Referee comment:

I am satisfied that the authors have addressed my comments systematically.

I have just one suggested minor revision, as follows.

In their response to my earlier review, the authors agreed with my comment about route establishment and variation. To avoid confusion on the new point they are making in this regard, I suggest the following rewording (changes in uppercase):

The corridors of the routes are often agreed upon between competing drivers or groups of service providers, and drivers of informal services typically adhere to these corridors to avoid conflict with competitors OR FINES FROM TRAFFIC POLICE. However, EVEN WHEN EXACT ROUTES ARE SPECIFIED IN OPERATING LICENSES, IN PRACTICE drivers (are in principle free to-DELETE) OFTEN adjust their route and stop locations, for example, to find more customers, circumvent traffic, or avoid police checkpoints when they receive relevant information from passing drivers.

Reason for rewording: in instances where route alignments are specified in operating licenses, it is not correct to say that the drivers are 'in principle free to adjust routes'.

Authors' response:

Due to the wide variety of possible operating schemes of informal transport, a short explanation as provided in the manuscript will always be highly simplified and cannot cover all aspects of these services. We thank the reviewer for their suggestion to clarify the explanation and have adopted the suggested change in the revised manuscript.